# Federated Multi-Task Learning

**Virginia Smith**
Stanford
smithv@stanford.edu

**Chao-Kai Chiang**[*]
USC
chaokaic@usc.edu

**Maziar Sanjabi**[*]
USC
maziarsanjabi@gmail.com

**Ameet Talwalkar**
CMU
talwalkar@cmu.edu

## Abstract

Federated learning poses new statistical and systems challenges in training machine learning models over distributed networks of devices. In this work, we show that multi-task learning is naturally suited to handle the statistical challenges of this setting, and propose a novel systems-aware optimization method, MOCHA, that is robust to practical systems issues. Our method and theory for the first time consider issues of high communication cost, stragglers, and fault tolerance for distributed multi-task learning. The resulting method achieves significant speedups compared to alternatives in the federated setting, as we demonstrate through simulations on real-world federated datasets.

## 1 Introduction

Mobile phones, wearable devices, and smart homes are just a few of the modern distributed networks generating massive amounts of data each day. Due to the growing storage and computational power of devices in these networks, it is increasingly attractive to store data locally and push more network computation to the edge. The nascent field of *federated learning* explores *training* statistical models directly on devices [37]. Examples of potential applications include: learning sentiment, semantic location, or activities of mobile phone users; predicting health events like low blood sugar or heart attack risk from wearable devices; or detecting burglaries within smart homes [3, 39, 42]. Following [25, 36, 26], we summarize the unique challenges of federated learning below.

1. **Statistical Challenges**: The aim in federated learning is to fit a model to data, $\{\mathbf{X}_1, \ldots, \mathbf{X}_m\}$, generated by $m$ distributed nodes. Each node, $t \in [m]$, collects data in a *non-IID* manner across the network, with data on each node being generated by a distinct distribution $\mathbf{X}_t \sim P_t$. The number of data points on each node, $n_t$, may also vary significantly, and there may be an underlying structure present that captures the relationship amongst nodes and their associated distributions.

2. **Systems Challenges**: There are typically a large number of nodes, $m$, in the network, and communication is often a significant bottleneck. Additionally, the storage, computational, and communication capacities of each node may differ due to variability in hardware (CPU, memory), network connection (3G, 4G, WiFi), and power (battery level). These systems challenges, compounded with unbalanced data and statistical heterogeneity, make issues such as stragglers and fault tolerance significantly more prevalent than in typical data center environments.

In this work, we propose a modeling approach that differs significantly from prior work on federated learning, where the aim thus far has been to train a single global model across the network [25, 36, 26]. Instead, we address statistical challenges in the federated setting by learning separate models for each node, $\{\mathbf{w}_1, \ldots, \mathbf{w}_m\}$. This can be naturally captured through a *multi-task learning (MTL)* framework, where the goal is to consider fitting separate but related models simultaneously [14, 2, 57, 28]. Unfortunately, current multi-task learning methods are not suited to handle the systems challenges that arise in federated learning, including high communication cost, stragglers, and fault tolerance. Addressing these challenges is therefore a key component of our work.

---

[*]Authors contributed equally.

## 1.1 Contributions

We make the following contributions. First, we show that MTL is a natural choice to handle statistical challenges in the federated setting. Second, we develop a novel method, MOCHA, to solve a general MTL problem. Our method generalizes the distributed optimization method CoCoA [22, 31] in order to address systems challenges associated with network size and node heterogeneity. Third, we provide convergence guarantees for MOCHA that carefully consider these unique systems challenges and provide insight into practical performance. Finally, we demonstrate the superior empirical performance of MOCHA with a new benchmarking suite of federated datasets.

## 2 Related Work

**Learning Beyond the Data Center.** Computing SQL-like queries across distributed, low-powered nodes is a decades-long area of research that has been explored under the purview of query processing in sensor networks, computing at the edge, and fog computing [32, 12, 33, 8, 18, 15]. Recent works have also considered training machine learning models centrally but serving and storing them locally, e.g., this is a common approach in mobile user modeling and personalization [27, 43, 44]. However, as the computational power of the nodes within distributed networks grows, it is possible to do even more work locally, which has led to recent interest in federated learning.[2] In contrast to our proposed approach, existing federated learning approaches [25, 36, 26, 37] aim to learn a single global model across the data.[3] This limits their ability to deal with non-IID data and structure amongst the nodes. These works also come without convergence guarantees, and have not addressed practical issues of stragglers or fault tolerance, which are important characteristics of the federated setting. The work proposed here is, to the best of our knowledge, the first federated learning framework to consider these challenges, theoretically and in practice.

**Multi-Task Learning.** In multi-task learning, the goal is to learn models for multiple related tasks simultaneously. While the MTL literature is extensive, most MTL modeling approaches can be broadly categorized into two groups based on how they capture relationships amongst tasks. The first (e.g., [14, 4, 11, 24]) assumes that a clustered, sparse, or low-rank structure between the tasks is known *a priori*. A second group instead assumes that the task relationships are not known beforehand and can be learned directly from the data (e.g., [21, 57, 16]). In this work, we focus our attention on this latter group, as task relationships may not be known beforehand in real-world settings. In comparison to learning a single global model, these MTL approaches can directly capture relationships amongst non-IID and unbalanced data, which makes them particularly well-suited for the statistical challenges of federated learning. We demonstrate this empirically on real-world federated datasets in Section 5. However, although MTL is a natural modeling choice to address the statistical challenges of federated learning, currently proposed methods for distributed MTL (discussed below) do not adequately address the systems challenges associated with federated learning.

**Distributed Multi-Task Learning.** Distributed multi-task learning is a relatively new area of research, in which the aim is to solve an MTL problem when data for each task is distributed over a network. While several recent works [1, 35, 54, 55] have considered the issue of distributed MTL training, the proposed methods do not allow for flexibility of communication versus computation. As a result, they are unable to efficiently handle concerns of fault tolerance and stragglers, the latter of which stems from both data and system heterogeneity. The works of [23] and [7] allow for asynchronous updates to help mitigate stragglers, but do not address fault tolerance. Moreover, [23] provides no convergence guarantees, and the convergence of [7] relies on a bounded delay assumption that is impractical for the federated setting, where delays may be significant and devices may drop out completely. Finally, [30] proposes a method and setup leveraging the distributed framework CoCoA [22, 31], which we show in Section 4 to be a special case of the more general approach in this work. However, the authors in [30] do not explore the federated setting, and their assumption that the same amount of work is done locally on each node is prohibitive in federated settings, where unbalance is common due to data and system variability.

# 3 Federated Multi-Task Learning

In federated learning, the aim is to learn a model over data that resides on, and has been generated by, $m$ distributed nodes. As a running example, consider learning the activities of mobile phone users in a cell network based on their individual sensor, text, or image data. Each node (phone), $t \in [m]$, may generate data via a distinct distribution, and so it is natural to fit separate models, $\{\mathbf{w}_1, \dots, \mathbf{w}_m\}$, to the distributed data—one for each local dataset. However, structure between models frequently exists (e.g., people may behave similarly when using their phones), and modeling these relationships via *multi-task learning* is a natural strategy to improve performance and boost the effective sample size for each node [10, 2, 5]. In this section, we suggest a general MTL framework for the federated setting, and propose a novel method, MOCHA, to handle the systems challenges of federated MTL.

## 3.1 General Multi-Task Learning Setup

Given data $\mathbf{X}_t \in \mathbb{R}^{d \times n_t}$ from $m$ nodes, multi-task learning fits separate weight vectors $\mathbf{w}_t \in \mathbb{R}^d$ to the data for each task (node) through arbitrary convex loss functions $\ell_t$ (e.g., the hinge loss for SVM models). Many MTL problems can be captured via the following general formulation:

$$\min_{\mathbf{W}, \mathbf{\Omega}} \left\{ \sum_{t=1}^{m} \sum_{i=1}^{n_t} \ell_t(\mathbf{w}_t^T \mathbf{x}_t^i, y_t^i) + \mathcal{R}(\mathbf{W}, \mathbf{\Omega}) \right\}, \tag{1}$$

where $\mathbf{W} := [\mathbf{w}_1, \dots, \mathbf{w}_m] \in \mathbb{R}^{d \times m}$ is a matrix whose $t$-th column is the weight vector for the $t$-th task. The matrix $\mathbf{\Omega} \in \mathbb{R}^{m \times m}$ models relationships amongst tasks, and is either known a priori or estimated while simultaneously learning task models. MTL problems differ based on their assumptions on $\mathcal{R}$, which takes $\mathbf{\Omega}$ as input and promotes some suitable structure amongst the tasks.

As an example, several popular MTL approaches assume that tasks form clusters based on whether or not they are related [14, 21, 57, 58]. This can be expressed via the following bi-convex formulation:

$$\mathcal{R}(\mathbf{W}, \mathbf{\Omega}) = \lambda_1 \operatorname{tr}(\mathbf{W} \mathbf{\Omega} \mathbf{W}^T) + \lambda_2 \|\mathbf{W}\|_F^2, \tag{2}$$

with constants $\lambda_1, \lambda_2 > 0$, and where the second term performs $L_2$ regularization on each local model. We use a similar formulation (14) in our experiments in Section 5, and provide details on other common classes of MTL models that can be formulated via (1) in Appendix B.

## 3.2 MOCHA: A Framework for Federated Multi-Task Learning

In the federated setting, the aim is to train statistical models directly on the edge, and thus we solve (1) while assuming that the data $\{\mathbf{X}_1, \dots, \mathbf{X}_m\}$ is distributed across $m$ nodes or devices. Before proposing our federated method for solving (1), we make the following observations:

- *Observation 1: In general,* (1) *is not jointly convex in* $\mathbf{W}$ *and* $\mathbf{\Omega}$*, and even in the cases where* (1) *is convex, solving for* $\mathbf{W}$ *and* $\mathbf{\Omega}$ *simultaneously can be difficult [5].*

- *Observation 2: When fixing* $\mathbf{\Omega}$*, updating* $\mathbf{W}$ *depends on both the data* $\mathbf{X}$*, which is distributed across the nodes, and the structure* $\mathbf{\Omega}$*, which is known centrally.*

- *Observation 3: When fixing* $\mathbf{W}$*, optimizing for* $\mathbf{\Omega}$ *only depends on* $\mathbf{W}$ *and not on the data* $\mathbf{X}$*.*

Based on these observations, it is natural to propose an alternating optimization approach to solve problem (1), in which at each iteration we fix either $\mathbf{W}$ or $\mathbf{\Omega}$ and optimize over the other, alternating until convergence is reached. Note that solving for $\mathbf{\Omega}$ is not dependent on the data and therefore can be computed centrally; as such, we defer to prior work for this step [58, 21, 57, 16]. In Appendix B, we discuss updates to $\mathbf{\Omega}$ for several common MTL models.

In this work, we focus on developing an efficient distributed optimization method for the $\mathbf{W}$ step. In traditional data center environments, the task of distributed training is a well-studied problem, and various communication-efficient frameworks have been recently proposed, including the state-of-the-art primal-dual COCOA framework [22, 31]. Although COCOA can be extended directly to update $\mathbf{W}$ in a distributed fashion across the nodes, it cannot handle the unique systems challenges of the federated environment, such as stragglers and fault tolerance, as discussed in Section 3.4. To this end, we extend COCOA and propose a new method, MOCHA, for federated multi-task learning. Our method is given in Algorithm 1 and described in detail in Sections 3.3 and 3.4.

**Algorithm 1** MOCHA: Federated Multi-Task Learning Framework
___
1:  **Input:** Data $\mathbf{X}_t$ from $t = 1, \ldots, m$ tasks, stored on one of $m$ nodes, and initial matrix $\boldsymbol{\Omega}_0$
2:  Starting point $\boldsymbol{\alpha}^{(0)} := \mathbf{0} \in \mathbb{R}^n$, $\mathbf{v}^{(0)} := \mathbf{0} \in \mathbb{R}^b$
3:  **for iterations** $i = 0, 1, \ldots$ **do**
4:      Set subproblem parameter $\sigma'$ and number of federated iterations, $H_i$
5:      **for iterations** $h = 0, 1, \cdots, H_i$ **do**
6:          **for tasks** $t \in \{1, 2, \ldots, m\}$ **in parallel over** $m$ **nodes do**
7:              call local solver, returning $\theta_t^h$-approximate solution $\Delta\boldsymbol{\alpha}_t$ of the local subproblem (4)
8:              update local variables $\boldsymbol{\alpha}_t \leftarrow \boldsymbol{\alpha}_t + \Delta\boldsymbol{\alpha}_t$
9:              return updates $\Delta\mathbf{v}_t := \mathbf{X}_t \Delta\boldsymbol{\alpha}_t$
10:         **reduce:** $\mathbf{v}_t \leftarrow \mathbf{v}_t + \Delta\mathbf{v}_t$
11:     Update $\boldsymbol{\Omega}$ centrally based on $\mathbf{w}(\boldsymbol{\alpha})$ for latest $\boldsymbol{\alpha}$
12: Central node computes $\mathbf{w} = \mathbf{w}(\boldsymbol{\alpha})$ based on the lastest $\boldsymbol{\alpha}$
13: **return:** $\mathbf{W} := [\mathbf{w}_1, \ldots, \mathbf{w}_m]$
___

## 3.3 Federated Update of W

To update $\mathbf{W}$ in the federated setting, we begin by extending works on distributed primal-dual optimization [22, 31, 30] to apply to the generalized multi-task framework (1). This involves deriving the appropriate dual formulation, subproblems, and problem parameters, as we detail below.

**Dual problem.** Considering the dual formulation of (1) will allow us to better separate the global problem into distributed subproblems for federated computation across the nodes. Let $n := \sum_{t=1}^{m} n_t$ and $\mathbf{X} := \text{Diag}(\mathbf{X}_1, \cdots, \mathbf{X}_m) \in \mathbb{R}^{md \times n}$. With $\boldsymbol{\Omega}$ fixed, the dual of problem (1), defined with respect to dual variables $\boldsymbol{\alpha} \in \mathbb{R}^n$, is given by:

$$\min_{\boldsymbol{\alpha}} \left\{ \mathcal{D}(\boldsymbol{\alpha}) := \sum_{t=1}^{m} \sum_{i=1}^{n_t} \ell_t^*(-\boldsymbol{\alpha}_t^i) + \mathcal{R}^*(\mathbf{X}\boldsymbol{\alpha}) \right\} , \qquad (3)$$

where $\ell_t^*$ and $\mathcal{R}^*$ are the conjugate dual functions of $\ell_t$ and $\mathcal{R}$, respectively, and $\boldsymbol{\alpha}_t^i$ is the dual variable for the data point $(\mathbf{x}_t^i, y_t^i)$. Note that $\mathcal{R}^*$ depends on $\boldsymbol{\Omega}$, but for the sake of simplicity, we have removed this in our notation. To derive distributed subproblems from this global dual, we make an assumption described below on the regularizer $\mathcal{R}$.

**Assumption 1.** *Given $\boldsymbol{\Omega}$, we assume that there exists a symmetric positive definite matrix $\mathbf{M} \in \mathbb{R}^{md \times md}$, depending on $\boldsymbol{\Omega}$, for which the function $\mathcal{R}$ is strongly convex with respect to $\mathbf{M}^{-1}$. Note that this corresponds to assuming that $\mathcal{R}^*$ will be smooth with respect to matrix $\mathbf{M}$.*

**Remark 1.** *We can reformulate the MTL regularizer in the form of $\bar{\mathcal{R}}(\mathbf{w}, \bar{\boldsymbol{\Omega}}) = \mathcal{R}(\mathbf{W}, \boldsymbol{\Omega})$, where $\mathbf{w} \in \mathbb{R}^{md}$ is a vector containing the columns of $\mathbf{W}$ and $\bar{\boldsymbol{\Omega}} := \boldsymbol{\Omega} \otimes \mathbf{I}_{d \times d} \in \mathbb{R}^{md \times md}$. For example, we can rewrite the regularizer in (2) as $\bar{\mathcal{R}}(\mathbf{w}, \bar{\boldsymbol{\Omega}}) = tr\big(\mathbf{w}^T(\lambda_1\bar{\boldsymbol{\Omega}} + \lambda_2\mathbf{I})\mathbf{w}\big)$. Writing the regularizer in this form, it is clear that it is strongly convex with respect to matrix $\mathbf{M}^{-1} = \lambda_1\bar{\boldsymbol{\Omega}} + \lambda_2\mathbf{I}$.*

**Data-local quadratic subproblems.** To solve (1) across distributed nodes, we define the following data-local subproblems, which are formed via a careful quadratic approximation of the dual problem (3) to separate computation across the nodes. These subproblems find updates $\Delta\boldsymbol{\alpha}_t \in \mathbb{R}^{n_t}$ to the dual variables in $\boldsymbol{\alpha}$ corresponding to a single node $t$, and only require accessing data which is available locally, i.e., $\mathbf{X}_t$ for node $t$. The $t$-th subproblem is given by:

$$\min_{\Delta\boldsymbol{\alpha}_t} \mathcal{G}_t^{\sigma'}(\Delta\boldsymbol{\alpha}_t; \mathbf{v}_t, \boldsymbol{\alpha}_t) := \sum_{i=1}^{n_t} \ell_t^*(-\boldsymbol{\alpha}_t^i - \Delta\boldsymbol{\alpha}_t^i) + \langle \mathbf{w}_t(\boldsymbol{\alpha}), \mathbf{X}_t\Delta\boldsymbol{\alpha}_t \rangle + \frac{\sigma'}{2} \|\mathbf{X}_t\Delta\boldsymbol{\alpha}_t\|_{\mathbf{M}_t}^2 + c(\boldsymbol{\alpha}) , \quad (4)$$

where $c(\boldsymbol{\alpha}) := \frac{1}{m}\mathcal{R}^*(\mathbf{X}\boldsymbol{\alpha})$, and $\mathbf{M}_t \in \mathbb{R}^{d \times d}$ is the $t$-th diagonal block of the symmetric positive definite matrix $\mathbf{M}$. Given dual variables $\boldsymbol{\alpha}$, corresponding primal variables can be found via $\mathbf{w}(\boldsymbol{\alpha}) = \nabla\mathcal{R}^*(\mathbf{X}\boldsymbol{\alpha})$, where $\mathbf{w}_t(\boldsymbol{\alpha})$ is the $t$-th block in the vector $\mathbf{w}(\boldsymbol{\alpha})$. Note that computing $\mathbf{w}(\boldsymbol{\alpha})$ requires the vector $\mathbf{v} = \mathbf{X}\boldsymbol{\alpha}$. The $t$-th block of $\mathbf{v}$, $\mathbf{v}_t \in \mathbb{R}^d$, is the only information that must be communicated between nodes at each iteration. Finally, $\sigma' > 0$ measures the difficulty of the data partitioning, and helps to relate progress made to the subproblems to the global dual problem. It can be easily selected based on $\mathbf{M}$ for many applications of interest; we provide details in Lemma 9 of the Appendix.

## 3.4 Practical Considerations

During MOCHA's federated update of $\mathbf{W}$, the central node requires a response from all workers before performing a synchronous update. In the federated setting, a naive execution of this communication protocol could introduce dramatic straggler effects due to node heterogeneity. To avoid stragglers, MOCHA provides the $t$-th node with the flexibility to *approximately solve* its subproblem $\mathcal{G}_t^{\sigma'}(\cdot)$, where the quality of the approximation is controled by a per-node parameter $\theta_t^h$. The following factors determine the quality of the $t$-th node's solution to its subproblem:

1. **Statistical challenges**, such as the size of $\mathbf{X}_t$ and the intrinsic difficulty of subproblem $\mathcal{G}_t^{\sigma'}(\cdot)$.
2. **Systems challenges**, such as the node's storage, computational, and communication capacities due to hardware (CPU, memory), network connection (3G, 4G, WiFi), and power (battery level).
3. A **global clock cycle** imposed by the central node specifying a deadline for receiving updates.

We define $\theta_t^h$ as a function of these factors, and assume that each node has a controller that may derive $\theta_t^h$ from the current clock cycle and statistical/systems setting. $\theta_t^h$ ranges from zero to one, where $\theta_t^h = 0$ indicates an exact solution to $\mathcal{G}_t^{\sigma'}(\cdot)$ and $\theta_t^h = 1$ indicates that node $t$ made no progress during iteration $h$ (which we refer to as a *dropped node*). For instance, a node may 'drop' if it runs out of battery, or if its network bandwidth deteriorates during iteration $h$ and it is thus unable to return its update within the current clock cycle. A formal definition of $\theta_t^h$ is provided in (5) of Section 4.

MOCHA mitigates stragglers by enabling the $t$-th node to define its own $\theta_t^h$. On every iteration $h$, the local updates that a node performs and sends in a clock cycle will yield a specific value for $\theta_t^h$. As discussed in Section 4, MOCHA is additionally robust to a small fraction of nodes periodically dropping and performing no local updates (i.e., $\theta_t^h := 1$) under suitable conditions, as defined in Assumption 2. In contrast, prior work of COCOA may suffer from severe straggler effects in federated settings, as it requires a *fixed $\theta_t^h = \theta$ across all nodes and all iterations* while still maintaining synchronous updates, and it does not allow for the case of dropped nodes ($\theta := 1$).

Finally, we note that asynchronous updating schemes are an alternative approach to mitigate stragglers. We do not consider these approaches in this work, in part due to the fact that the bounded-delay assumptions associated with most asynchronous schemes limit fault tolerance. However, it would be interesting to further explore the differences and connections between asynchronous methods and approximation-based, synchronous methods like MOCHA in future work.

## 4 Convergence Analysis

MOCHA is based on a bi-convex alternating approach, which is guaranteed to converge [17, 45] to a stationary solution of problem (1). In the case where this problem is jointly convex with respect to $\mathbf{W}$ and $\mathbf{\Omega}$, such a solution is also optimal. In the rest of this section, we therefore focus on the convergence of solving the $\mathbf{W}$ update of MOCHA in the federated setting. Following the discussion in Section 3.4, we first introduce the following per-node, per-round approximation parameter.

**Definition 1** (Per-Node-Per-Iteration-Approximation Parameter). At each iteration $h$, we define the accuracy level of the solution calculated by node $t$ to its subproblem (4) as:

$$\theta_t^h := \frac{\mathcal{G}_t^{\sigma'}(\Delta\boldsymbol{\alpha}_t^{(h)}; \mathbf{v}^{(h)}, \boldsymbol{\alpha}_t^{(h)}) - \mathcal{G}_t^{\sigma'}(\Delta\boldsymbol{\alpha}_t^{\star}; \mathbf{v}^{(h)}, \boldsymbol{\alpha}_t^{(h)})}{\mathcal{G}_t^{\sigma'}(\mathbf{0}; \mathbf{v}^{(h)}, \boldsymbol{\alpha}_t^{(h)}) - \mathcal{G}_t^{\sigma'}(\Delta\boldsymbol{\alpha}_t^{\star}; \mathbf{v}^{(h)}, \boldsymbol{\alpha}_t^{(h)})}, \qquad (5)$$

where $\Delta\boldsymbol{\alpha}_t^{\star}$ is the minimizer of subproblem $\mathcal{G}_t^{\sigma'}(\cdot\,; \mathbf{v}^{(h)}, \boldsymbol{\alpha}_t^{(h)})$. We allow this value to vary between $[0, 1]$, with $\theta_t^h := 1$ meaning that no updates to subproblem $\mathcal{G}_t^{\sigma'}$ are made by node $t$ at iteration $h$.

While the flexible per-node, per-iteration approximation parameter $\theta_t^h$ in (5) allows the consideration of stragglers and fault tolerance, these additional degrees of freedom also pose new challenges in providing convergence guarantees for MOCHA. We introduce the following assumption on $\theta_t^h$ to provide our convergence guarantees.

**Assumption 2.** *Let $\mathcal{H}_h := (\boldsymbol{\alpha}^{(h)}, \boldsymbol{\alpha}^{(h-1)}, \cdots, \boldsymbol{\alpha}^{(1)})$ be the dual vector history until the beginning of iteration $h$, and define $\Theta_t^h := \mathbb{E}[\theta_t^h | \mathcal{H}_h]$. For all tasks $t$ and all iterations $h$, we assume $p_t^h := \mathbb{P}[\theta_t^h = 1] \le p_{\max} < 1$ and $\hat{\Theta}_t^h := \mathbb{E}[\theta_t^h | \mathcal{H}_h, \theta_t^h < 1] \le \Theta_{\max} < 1$.*

This assumption states that at each iteration, the *probability* of a node sending a result is non-zero, and that the quality of the returned result is, on average, better than the previous iterate. Compared to [49, 30] which assumes $\theta_t^h = \theta < 1$, our assumption is significantly less restrictive and better models the federated setting, where nodes are unreliable and may periodically drop out.

Using Assumption 2, we derive the following theorem, which characterizes the convergence of the federated update of MOCHA in finite horizon when the losses $\ell_t$ in (1) are smooth.

**Theorem 1.** *Assume that the losses $\ell_t$ are $(1/\mu)$-smooth. Then, under Assumptions 1 and 2, there exists a constant $s \in (0, 1]$ such that for any given convergence target $\epsilon_{\mathcal{D}}$, choosing $H$ such that*

$$H \geq \frac{1}{(1 - \bar{\Theta})s} \log \frac{n}{\epsilon_{\mathcal{D}}} \,, \tag{6}$$

*will satisfy $\mathbb{E}[\mathcal{D}(\boldsymbol{\alpha}^{(H)}) - \mathcal{D}(\boldsymbol{\alpha}^{\star})] \leq \epsilon_{\mathcal{D}}$ .*

Here, $\bar{\Theta} := p_{\max} + (1 - p_{\max})\Theta_{\max} < 1$. While Theorem 1 is concerned with finite horizon convergence, it is possible to get asymptotic convergence results, i.e., $H \to \infty$, with milder assumptions on the stragglers; see Corollary 8 in the Appendix for details.

When the loss functions are non-smooth, e.g., the hinge loss for SVM models, we provide the following sub-linear convergence for $L$-Lipschitz losses.

**Theorem 2.** *If the loss functions $\ell_t$ are $L$-Lipschitz, then there exists a constant $\sigma$, defined in (24), such that for any given $\epsilon_{\mathcal{D}} > 0$, if we choose*

$$H \geq H_0 + \left\lceil \frac{2}{(1 - \bar{\Theta})} \max\left(1, \frac{2L^2 \sigma \sigma'}{n^2 \epsilon_{\mathcal{D}}}\right) \right\rceil, \tag{7}$$

$$\text{with } H_0 \geq \left\lceil h_0 + \frac{16L^2 \sigma \sigma'}{(1 - \bar{\Theta})n^2 \epsilon_{\mathcal{D}}} \right\rceil, h_0 = \left\lceil 1 + \frac{1}{(1 - \bar{\Theta})} \log\left(\frac{2n^2(D(\boldsymbol{\alpha}^{\star}) - D(\boldsymbol{\alpha}^0))}{4L^2 \sigma \sigma'}\right) \right\rceil_+,$$

*then $\bar{\boldsymbol{\alpha}} := \frac{1}{H - H0} \sum_{h=H_0+1}^{H} \boldsymbol{\alpha}^{(h)}$ will satisfy $\mathbb{E}[\mathcal{D}(\bar{\boldsymbol{\alpha}}) - \mathcal{D}(\boldsymbol{\alpha}^{\star})] \leq \epsilon_{\mathcal{D}}$ .*

These theorems guarantee that MOCHA will converge in the federated setting, under mild assumptions on stragglers and capabilities of the nodes. While these results consider convergence in terms of the dual, we show that they hold analogously for the duality gap. We provide all proofs in Appendix C.

**Remark 2.** *Following from the discussion in Section 3.4, our method and theory generalize the results in [22, 31]. In the limiting case that all $\theta_t^h$ are identical, our results extend the results of COCOA to the multi-task framework described in (1).*

**Remark 3.** *Note that the methods in [22, 31] have an aggregation parameter $\gamma \in (0, 1]$. Though we prove our results for a general $\gamma$, we simplify the method and results here by setting $\gamma := 1$, which has been shown to have the best performance, both theoretically and empirically [31].*

## 5 Simulations

In this section we validate the empirical performance of MOCHA. First, we introduce a benchmarking suite of real-world federated datasets and show that multi-task learning is well-suited to handle the statistical challenges of the federated setting. Next, we demonstrate MOCHA's ability to handle stragglers, both from statistical and systems heterogeneity. Finally, we explore the performance of MOCHA when devices periodically drop out. Our code is available at: `github.com/gingsmith/fmtl`.

### 5.1 Federated Datasets

In our simulations, we use several real-world datasets that have been generated in federated settings. We provide additional details in the Appendix, including information about data sizes, $n_t$.

- **Google Glass (GLEAM)**[4]: This dataset consists of two hours of high resolution sensor data collected from 38 participants wearing Google Glass for the purpose of activity recognition. Following [41], we featurize the raw accelerometer, gyroscope, and magnetometer data into 180 statistical, spectral, and temporal features. We model each participant as a separate task, and predict between eating and other activities (e.g., walking, talking, drinking).

- **Human Activity Recognition**[5]: Mobile phone accelerometer and gyroscope data collected from 30 individuals, performing one of six activities: {*walking, walking-upstairs, walking-downstairs, sitting, standing, lying-down*}. We use the provided 561-length feature vectors of time and frequency domain variables generated for each instance [3]. We model each individual as a separate task and predict between sitting and the other activities.

- **Vehicle Sensor**[6]: Acoustic, seismic, and infrared sensor data collected from a distributed network of 23 sensors, deployed with the aim of classifying vehicles driving by a segment of road [13]. Each instance is described by 50 acoustic and 50 seismic features. We model each sensor as a separate task and predict between AAV-type and DW-type vehicles.

## 5.2 Multi-Task Learning for the Federated Setting

We demonstrate the benefits of multi-task learning for the federated setting by comparing the error rates of a multi-task model to that of a fully local model (i.e., learning a model for each task separately) and a fully global model (i.e., combining the data from all tasks and learning one single model). Work on federated learning thus far has been limited to the study of fully global models [25, 36, 26].

We use a cluster-regularized multi-task model [57], as described in Section 3.1. For each dataset from Section 5.1, we randomly split the data into 75% training and 25% testing, and learn multi-task, local, and global support vector machine models, selecting the best regularization parameter, $\lambda \in \{$1e-5, 1e-4, 1e-3, 1e-2, 0.1, 1, 10$\}$, for each model using 5-fold cross-validation. We repeat this process 10 times and report the average prediction error across tasks, averaged across these 10 trials.

Table 1: Average prediction error: Means and standard errors over 10 random shuffles.

| **Model** | Human Activity | Google Glass | Vehicle Sensor |
|---|---|---|---|
| Global | 2.23 (0.30) | 5.34 (0.26) | 13.4 (0.26) |
| Local | 1.34 (0.21) | 4.92 (0.26) | 7.81 (0.13) |
| MTL | **0.46 (0.11)** | **2.02 (0.15)** | **6.59 (0.21)** |

In Table 1, we see that for each dataset, multi-task learning significantly outperforms the other models in terms of achieving the lowest average error across tasks. The global model, as proposed in [25, 36, 26] performs the worst, particularly for the Human Activity and Vehicle Sensor datasets. Although the datasets are already somewhat unbalanced, we note that a global modeling approach may benefit tasks with a very small number of instances, as information can be shared across tasks. For this reason, we additionally explore the performance of global, local, and multi-task modeling for highly skewed data in Table 4 of the Appendix. Although the performance of the global model improves slightly relative to local modeling in this setting, the global model still performs the worst for the majority of the datasets, and MTL still significantly outperforms both global and local approaches.

## 5.3 Straggler Avoidance

Two challenges that are prevalent in federated learning are stragglers and high communication. Stragglers can occur when a subset of the devices take much longer than others to perform local updates, which can be caused either by statistical or systems heterogeneity. Communication can also exacerbate poor performance, as it can be slower than computation by many orders of magnitude in typical cellular or wireless networks [52, 20, 48, 9, 38]. In our experiments below, we simulate the time needed to run each method by tracking the operations and communication complexities, and scaling the communication cost relative to computation by one, two, or three orders of magnitude, respectively. These numbers correspond roughly to the clock rate vs. network bandwidth/latency (see, e.g., [52]) for modern cellular and wireless networks. Details are provided in Appendix E.

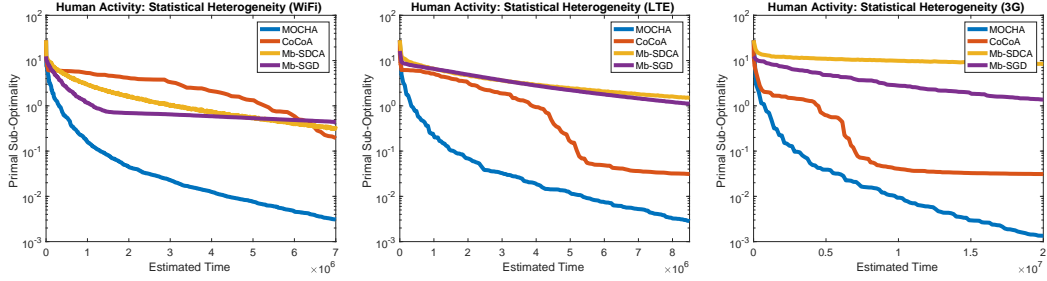

Figure 1: The performance of MOCHA compared to other distributed methods for the $\mathbf{W}$ update of (1). While increasing communication tends to *decrease* the performance of the mini-batch methods, MOCHA performs well in high communication settings. In all settings, MOCHA with varied approximation values, $\Theta_t^h$, performs better than without (i.e., naively generalizing CoCoA), as it avoids stragglers from statistical heterogeneity.

**Statistical Heterogeneity.** We explore the effect of statistical heterogeneity on stragglers for various methods and communication regimes (3G, LTE, WiFi). For a fixed communication network, we compare MOCHA to CoCoA, which has a single $\theta$ parameter, and to mini-batch stochastic gradient descent (Mb-SGD) and mini-batch stochastic dual coordinate ascent (Mb-SDCA), which have limited communication flexibility depending on the batch size. We tune all compared methods for best performance, as we detail in Appendix E. In Figure 1, we see that while the performance degrades for mini-batch methods in high communication regimes, MOCHA and CoCoA are robust to high communication. However, CoCoA is significantly affected by stragglers—because $\theta$ is fixed across nodes and rounds, difficult subproblems adversely impact convergence. In contrast, MOCHA performs well regardless of communication cost and is robust to statistical heterogeneity.

**Systems Heterogeneity.** MOCHA is also equipped to handle heterogeneity from changing systems environments, such as battery power, memory, or network connection, as we show in Figure 2. In particular, we simulate systems heterogeneity by randomly choosing the number of local iterations for MOCHA or the mini-batch size for mini-batch methods, between $10\%$ and $100\%$ of the minimum number of local data points for high variability environments, to between $90\%$ and $100\%$ for low variability (see Appendix E for full details). We do not vary the performance of CoCoA, as the impact from statistical heterogeneity alone significantly reduces performance. However, adding systems heterogeneity would reduce performance even further, as the maximum $\theta$ value across all nodes would only increase if additional systems challenges were introduced.

## 5.4 Tolerance to Dropped Nodes

Finally, we explore the effect of nodes dropping on the performance of MOCHA. We do not draw comparisons to other methods, as to the best of our knowledge, no other methods for distributed multi-task learning directly address fault tolerance. In MOCHA, we incorporate this setting by allowing $\theta_t^h := 1$, as explored theoretically in Section 4. In Figure 3, we look at the performance of MOCHA, either for one fixed $\mathbf{W}$ update, or running the entire MOCHA method, as the probability that nodes drop at each iteration ($p_t^h$ in Assumption 2) increases. We see that the performance of MOCHA is robust to relatively high values of $p_t^h$, both during a single update of $\mathbf{W}$ and in how this affects the performance of the overall method. However, as intuition would suggest, if one of the nodes *never* sends updates (i.e., $p_1^h := 1$ for all $h$, green dotted line), the method does not converge to the correct solution. This provides validation for our Assumption 2.

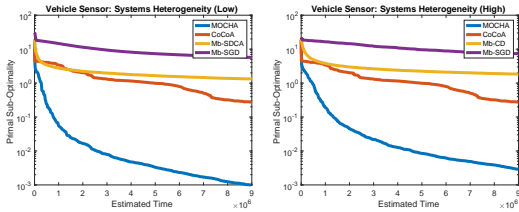

Figure 2: MOCHA can handle variability from systems heterogeneity.

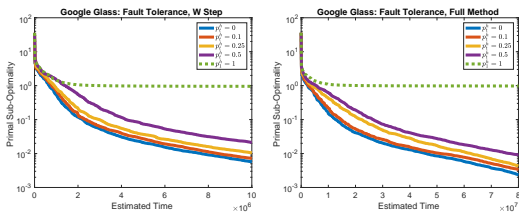

Figure 3: The performance of MOCHA is robust to nodes periodically dropping out (fault tolerance).

# 6 Discussion

To address the statistical and systems challenges of the burgeoning federated learning setting, we have presented MOCHA, a novel systems-aware optimization framework for federated multi-task learning. Our method and theory for the first time consider issues of high communication cost, stragglers, and fault tolerance for multi-task learning in the federated environment. While MOCHA does not apply to non-convex deep learning models in its current form, we note that there may be natural connections between this approach and "convexified" deep learning models [6, 34, 51, 56] in the context of kernelized federated multi-task learning.

## Acknowledgements

We thank Brendan McMahan, Chloé Kiddon, Jakub Konečný, Evan Sparks, Xinghao Pan, Lisha Li, and Hang Qi for valuable discussions and feedback.

## Footnotes

[2]The term *on-device learning* has been used to describe both the task of model training and of model serving. Due to the ambiguity of this phrase, we exclusively use the term federated learning.

[3]While not the focus of our work, we note privacy is an important concern in the federated setting, and that the privacy benefits associated with global federated learning (as discussed in [36]) also apply to our approach.

[4] `http://www.skleinberg.org/data/GLEAM.tar.gz`

[5]https://archive.ics.uci.edu/ml/datasets/Human+Activity+Recognition+Using+Smartphones

[6]http://www.ecs.umass.edu/~mduarte/Software.html

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
