[Supplementary Material]

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

# A  Preliminaries

**Notation.**  We use $\mathbf{I}_{d \times d}$ to represent an identity matrix of size $d \times d$. When the context allows, we use the notation $\mathbf{I}$ to denote an identity matrix of an appropriate size. We also use $\otimes$ to denote the Kronecker product between two matrices.

**Definition 2** (Matrix norm)**.**  Given a symmetric positive definite matrix $\mathbf{M}$, the norm of $\mathbf{u}$ with respect to $\mathbf{M}$ is given by $\|\mathbf{u}\|_{\mathbf{M}} := \sqrt{\mathbf{u}^T \mathbf{M} \mathbf{u}}$ .

**Definition 3** ($L$-smooth)**.**  A convex function $f$ is $L$-smooth with respect to $\mathbf{M}$ if

$$f(\mathbf{u}) \leq f(\mathbf{v}) + \langle \nabla f(\mathbf{v}), \mathbf{u} - \mathbf{v} \rangle + \frac{L}{2} \|\mathbf{u} - \mathbf{v}\|_{\mathbf{M}}^2 \qquad \forall \mathbf{u}, \mathbf{v} \,. \tag{8}$$

If $\mathbf{M} = \mathbf{I}$ then, we simply say $f$ is $L$-smooth.

**Definition 4** ($\tau$-strongly convex)**.**  A function $f$ is $\tau$-strongly convex with respect to $\mathbf{M}$ if

$$f(\mathbf{u}) \geq f(\mathbf{v}) + \langle \mathbf{z}, \mathbf{u} - \mathbf{v} \rangle + \frac{\tau}{2} \|\mathbf{u} - \mathbf{v}\|_{\mathbf{M}}^2 \qquad \forall \mathbf{u}, \mathbf{v}, \, \mathbf{z} \in \partial f(\mathbf{v}) \,, \tag{9}$$

where $\partial f(\mathbf{v})$ is the set of sub-differentials of function $f$ at $\mathbf{v}$. If $\mathbf{M} = \mathbf{I}$ then, we simply say $f$ is $\tau$-strongly convex.

**Definition 5.**  The function $f$ is called $L$-Lipchitz if for any $\mathbf{x}$ and $\mathbf{y}$ in its domain

$$|f(\mathbf{x}) - f(\mathbf{y})| \leq L \|\mathbf{x} - \mathbf{y}\|_2 \,. \tag{10}$$

If a function $f$ is $L$-Lipchitz then its dual will be $L$-bounded, i.e., for any $\boldsymbol{\alpha}$ such that $\|\boldsymbol{\alpha}\|_2 > L$, then $f^*(\boldsymbol{\alpha}) = +\infty$.

# B  Multi-Task Learning

In this section, we summarize several popular multi-task learning formulations that can be written in the form of (1) and can therefore be addressed by our framework, MOCHA. While the $\mathbf{W}$ update is discussed in Section 3, we provide details here on how to solve the $\Omega$ update for these formulations.

## B.1  Multi-Task Learning Formulations

**MTL with cluster structure.**  In MTL models that assume a cluster structure, the weight vectors for each task, $\mathbf{w}_t$, are assumed to 'close' according to some metric to other weight vectors from tasks in the same cluster. This idea goes back to mean-regularized MTL [14], which assumes that all the tasks form one cluster, and that the weight vectors are close to their mean. Such a regularizer could be formulated in the form of (1) by choosing $\Omega = (\mathbf{I}_{m \times m} - \frac{1}{m} \mathbf{1} \mathbf{1}^T)^2$, where $\mathbf{I}_{m \times m}$ is the identity matrix of size $m \times m$ and $\mathbf{1}_m$ represents a vector of all ones with size $m$. In this case, we set $\mathcal{R}$ to be

$$\mathcal{R}(\mathbf{W}, \Omega) = \lambda_1 \operatorname{tr}(\mathbf{W} \Omega \mathbf{W}^T) + \lambda_2 \|\mathbf{W}\|_F^2 \,, \tag{11}$$

where $\lambda_1, \lambda_2 > 0$ are parameters. Note that in this formulation, the structural dependence matrix $\Omega$ is known a-priori. However, it is natural to assume multiple clusters exist, and to learn this clustering structure directly from the data [58]. For such a model, the problem formulation is non-convex if a perfect clustering structure is imposed [58, 21]. However, by performing a convex relaxation, the following regularizer is obtained [58, 21]

$$\mathcal{R}(\mathbf{W}, \Omega) = \lambda \operatorname{tr}(\mathbf{W}(\eta \mathbf{I} + \Omega)^{-1} \mathbf{W}^T), \; \Omega \in \mathcal{Q} = \left\{ \mathbf{Q} \mid \mathbf{Q} \succeq \mathbf{0}, \; \operatorname{tr}(\mathbf{Q}) = k, \; \mathbf{Q} \preceq \mathbf{I} \right\}, \tag{12}$$

where $\lambda$ and $\eta$ are regularization parameters, $k$ is the number of clusters, and $\Omega$ defines the clustering structure.

**MTL with probabilistic priors.**  Another set of MTL models that can be realized by our framework enforce structure by putting probabilistic priors on the dependence among the columns of $\mathbf{W}$. For example, in [57] it is assumed that the weight matrix $\mathbf{W}$ has a prior distribution of the form:

$$\mathbf{W} \sim \left( \prod_{i=1}^{m} \mathcal{N}(\mathbf{0}, \sigma^2 \mathbf{I}) \right) \mathcal{M} \mathcal{N}(\mathbf{0}, \mathbf{I}_{d \times d} \otimes \Omega) \,, \tag{13}$$

where $\mathcal{N}(\mathbf{0}, \sigma^2\mathbf{I})$ denotes the normal distribution with mean $\mathbf{0}$ and covariance $\sigma^2\mathbf{I}$, and $\mathcal{MN}(\mathbf{0}, \mathbf{I}_{d\times d} \otimes \boldsymbol{\Omega})$ denotes the matrix normal distribution with mean $\mathbf{0}$, row covariance $\mathbf{I}_{d\times d}$, and column covariance $\boldsymbol{\Omega}$. This prior generates a regularizer of the following form [57]:

$$\mathcal{R}(\mathbf{W}, \boldsymbol{\Omega}) = \lambda \left( \frac{1}{\sigma^2} \|\mathbf{W}\|^2 + \operatorname{tr}(\mathbf{W}\boldsymbol{\Omega}^{-1}\mathbf{W}^T) + d \, \log |\boldsymbol{\Omega}| \right), \ \lambda > 0\,.$$

Unfortunately, such a regularizer is non-convex with respect to $\boldsymbol{\Omega}$ due to the concavity of $\log |\boldsymbol{\Omega}|$. To obtain a jointly convex formulation in $\boldsymbol{\Omega}$ and $\mathbf{W}$, the authors in [57] propose omitting $\log |\boldsymbol{\Omega}|$ and controlling the complexity of $\boldsymbol{\Omega}$ by adding a constraint on $\operatorname{tr}(\boldsymbol{\Omega})$:

$$\mathcal{R}(\mathbf{W}, \boldsymbol{\Omega}) = \lambda \left( \frac{1}{\sigma^2} \|\mathbf{W}\|^2 + \operatorname{tr}(\mathbf{W}\boldsymbol{\Omega}^{-1}\mathbf{W}^T) \right), \ \boldsymbol{\Omega} \in \mathcal{Q} = \left\{ \mathbf{Q} \mid \mathbf{Q} \succeq \mathbf{0}, \ \operatorname{tr}(\mathbf{Q}) = 1 \right\}. \quad (14)$$

It is worth noting that unlike the clustered MTL formulations, such as (2), the probabilistic formulation in (14) can model both positive and negative relationships among the tasks through the covariance matrix.

**MTL with graphical models.** Another way of modeling task relationships is through the precision matrix. This is popular in graphical models literature [29] because it encodes conditional independence among variables. In other words, if we denote the precision matrix among tasks in matrix variate Gaussian prior with $\boldsymbol{\Omega}$, then $\boldsymbol{\Omega}_{i,j} = 0$ if and only if tasks weights $\mathbf{w}_i$ and $\mathbf{w}_j$ are independent given the rest of the task weights [16]. Therefore, assuming sparsity in the structure among the tasks translates to sparsity in matrix $\boldsymbol{\Omega}$. As a result, we can formulate a sparsity-promoting regularizer by:

$$\mathcal{R}(\mathbf{W}, \boldsymbol{\Omega}) = \lambda \left( \frac{1}{\sigma^2} \|\mathbf{W}\|^2 + \operatorname{tr}(\mathbf{W}\boldsymbol{\Omega}\mathbf{W}^T) - d \, \log |\boldsymbol{\Omega}| \right) + \lambda_1 \|\mathbf{W}\|_1 + \lambda_2 \|\boldsymbol{\Omega}\|_1\,, \quad (15)$$

where $\lambda_1, \lambda_2 \geq 0$ control the sparsity of $\mathbf{W}$ and $\boldsymbol{\Omega}$ respectively [16]. It is worth noting that although this problem is jointly non-convex in $\mathbf{W}$ and $\boldsymbol{\Omega}$, it is bi-convex.

## B.2  Strong Convexity of MTL Regularizers

Recall that in Assumption 1, we presumed that the vectorized formulation of the MTL regularizer is strongly convex with respect to a matrix $\mathbf{M}^{-1}$. In this subsection we discuss the choice of matrix $\mathbf{M}$ for the widely-used MTL formulations introduced in Section B.1.

Using the notation from Remark 1 for the clustered MTL formulation (11), it is easy to see that $\bar{\mathcal{R}}(\mathbf{w}, \bar{\boldsymbol{\Omega}}) = \lambda_1 \mathbf{w}^T \bar{\boldsymbol{\Omega}} \mathbf{w} + \lambda_2 \|\mathbf{w}\|_2^2$, where $\bar{\boldsymbol{\Omega}} := \boldsymbol{\Omega} \otimes \mathbf{I}_{d\times d}$. As a result, it is clear that $\bar{\mathcal{R}}(\mathbf{w}, \bar{\boldsymbol{\Omega}})$ is 1-strongly convex with respect to $\mathbf{M}^{-1} = \lambda_1 \bar{\boldsymbol{\Omega}} + \lambda_2 \mathbf{I}_{md\times md}$.

Using a similar reasoning, it is easy to see that the matrix $\mathbf{M}$ can be chosen as $\lambda^{-1}(\eta\mathbf{I} + \bar{\boldsymbol{\Omega}})$, $\lambda^{-1}(\frac{1}{\sigma^2}\mathbf{I} + \bar{\boldsymbol{\Omega}}^{-1})^{-1}$ and $\lambda^{-1}(\frac{1}{\sigma^2}\mathbf{I} + \bar{\boldsymbol{\Omega}})^{-1}$ for (12), (14) and (15) respectively.

## B.3  Optimizing $\Omega$ in MTL Formulations

In this section, we briefly cover approaches to update $\boldsymbol{\Omega}$ in the MTL formulations introduced in Section B.1. First, it is clear that (2) does not require any updates to $\boldsymbol{\Omega}$, as it is assumed to be fixed. In (12), it can be shown [58, 21] that the optimal solution for $\boldsymbol{\Omega}$ has the same column space as the rows of $\mathbf{W}$. Therefore, the problem boils down to solving a simple convex optimization problem over the eigenvalues of $\boldsymbol{\Omega}$; see [58, 21] for details. Although outside the scope of this paper, we note that the bottleneck of this approach to finding $\boldsymbol{\Omega}$ is computing the SVD of $\mathbf{W}$, which can be a challenging problem when $m$ is large. In the probabilistic model of (14), the $\boldsymbol{\Omega}$ update is given in [57] by $(\mathbf{W}^T\mathbf{W})^{\frac{1}{2}}$, which requires computing the eigenvalue decomposition of $\mathbf{W}^T\mathbf{W}$. For the graphical model formulation, the problem of solving for $\boldsymbol{\Omega}$ is called sparse precision matrix estimation or graphical lasso [16]. This is a well-studied problem, and many scalable algorithms have been proposed to solve it [53, 16, 19].

### B.3.1  Reducing the Size of $\Omega$ by Sharing Tasks

One interesting aspect of MOCHA is that the method can be easily modified to accommodate the sharing of tasks among the nodes without any change to the local solvers. This property helps the central node to reduce the size of $\boldsymbol{\Omega}$ and the complexity of its update with minimal changes to the whole system. The following remark highlights this capability.

**Remark 4.** MOCHA *can be modified to solve problems when there are tasks that are shared among nodes. In this case, each node still solves a data local sub-problem based on its own data for the task, but the central node needs to do an additional aggregation step to add the results for all the nodes that share the data of each task. This reduces the size of matrix $\mathbf{\Omega}$ and simplifies its update.*

## C  Convergence Analysis

**Notation.**   In the rest of this section we use the superscript $(h)$ or $h$ to denote the corresponding variable at iteration $(h)$ of the federated update in MOCHA. When context allows, we drop the superscript to simplify notation.

In order to provide a general convergence analysis, similar to the ones provided in [22, 31, 49], we assume an aggregation parameter $\gamma \in (0, 1]$ in this section. With such an aggregation parameter, the updates in each federated iteration would be $\boldsymbol{\alpha}_t \leftarrow \boldsymbol{\alpha}_t + \gamma \Delta \boldsymbol{\alpha}_t$ and $\mathbf{v}_t \leftarrow \mathbf{v}_t + \gamma \Delta \mathbf{v}_t$. For a more detailed discussion on the role of aggregation parameter, see Appendix D.1. Note that in Algorithm 1, MOCHA is presented assuming $\gamma = 1$ for simplicity.

Before proving our convergence guarantees, we provide several useful definitions and key lemmas.

**Definition 6.**  For each task $t$, define

$$\sigma_t := \max_{\boldsymbol{\alpha} \in \mathbb{R}^{n_t}} \frac{\|\mathbf{X}_t \boldsymbol{\alpha}\|_{M_t}^2}{\|\boldsymbol{\alpha}\|^2} \quad \text{and} \quad \sigma_{\max} := \max_{t \in [m]} \sigma_t. \tag{16}$$

**Definition 7.**  For any $\boldsymbol{\alpha}$, define the duality gap as

$$G(\boldsymbol{\alpha}) := \mathcal{D}(\boldsymbol{\alpha}) - (-\mathcal{P}(\mathbf{W}(\boldsymbol{\alpha}))), \tag{17}$$

where $\mathcal{P}(\mathbf{W}) := \sum_{t=1}^{m} \sum_{i=1}^{n_t} \ell_t(\mathbf{w}_t^T \mathbf{x}_t^i, y_t^i) + \mathcal{R}(\mathbf{W}, \mathbf{\Omega})$ as in (1).

The following lemma uses Assumption 2 to bound the average performance of $\theta_t^h$, which is crucial in providing global convergence guarantees for MOCHA.

**Lemma 3.**  *Under Assumption 2,* $\Theta_t^h \leq \bar{\Theta} = p_{\max} + (1 - p_{\max})\Theta_{\max} < 1.$

*Proof.*  Recalling the definitions $p_t^h := \mathbb{P}[\theta_t^h = 1]$ and $\hat{\Theta}_t^h := \mathbb{E}[\theta_t^h | \theta_t^h < 1, \mathcal{H}_h]$, we have

$$\begin{aligned} \Theta_t^h &= \mathbb{E}[\theta_t^h | \mathcal{H}_h] \\ &= \mathbb{P}[\theta_t^h = 1] \cdot \mathbb{E}[\theta_t^h | \theta_t^h = 1, \mathcal{H}_h] + (1 - \mathbb{P}[\theta_t^h < 1]) \cdot \mathbb{E}[\theta_t^h | \theta_t^h < 1, \mathcal{H}_h] \\ &= p_t^h \cdot 1 + (1 - p_t^h) \cdot \hat{\Theta}_t^h \leq \bar{\Theta} < 1, \end{aligned}$$

where the last inequality is due to Assumption 2, and the fact that $\hat{\Theta}_t^h < 1$ by definition.  $\square$

The next key lemma bounds the dual objective of an iterate based on the dual objective of the previous iterate and the objectives of local subproblems.

**Lemma 4.**  *For any* $\boldsymbol{\alpha}, \Delta \boldsymbol{\alpha} \in \mathbb{R}^n$ *and* $\gamma \in (0, 1]$ *if* $\sigma'$ *satisfies (28), then*

$$\mathcal{D}(\boldsymbol{\alpha} + \gamma \Delta \boldsymbol{\alpha}) \leq (1 - \gamma)\mathcal{D}(\boldsymbol{\alpha}) + \gamma \sum_{t=1}^{m} \mathcal{G}_t^{\sigma'}(\Delta \boldsymbol{\alpha}_t; \mathbf{v}, \boldsymbol{\alpha}_t). \tag{18}$$

*Proof.*  The proof of this lemma is similar to [49, Lemma 1] and follows from the definition of local sub-problems, smoothness of $\mathcal{R}^*$ and the choice of $\sigma'$ in (28).  $\square$

Recall that if the functions $\ell_t$ are $(1/\mu)$-smooth, their conjugates $\ell_t^*$ will be $\mu$-strongly convex. The lemma below provides a bound on the amount of improvement in dual objective in each iteration.

**Lemma 5.**  *If the functions* $\ell_t^*$ *are* $\mu$-strongly convex for some $\mu \geq 0$. Then, for any $s \in [0, 1]$.

$$\mathbb{E}[\mathcal{D}(\boldsymbol{\alpha}^{(h)}) - \mathcal{D}(\boldsymbol{\alpha}^{(h+1)})|\mathcal{H}_h] \geq \gamma \sum_{t=1}^{m} (1 - \bar{\Theta}) \left( sG_t(\boldsymbol{\alpha}^{(h)}) - \frac{\sigma' s^2}{2} J_t \right), \tag{19}$$

*where*

$$G_t(\boldsymbol{\alpha}) := \sum_{i=1}^{n_t} \left[ \ell_t^*(-\boldsymbol{\alpha}_t^i) + \ell_t(\mathbf{w}_t(\boldsymbol{\alpha})^\top \mathbf{x}_t^i, y_t^i) + \boldsymbol{\alpha}_t^i \mathbf{w}_t(\boldsymbol{\alpha})^\top \mathbf{x}_t^i \right] , \tag{20}$$

$$J_t := -\frac{\mu(1-s)}{\sigma' s} \|(\mathbf{u}_t - \boldsymbol{\alpha}_t^{(h)})\|^2 + \|\mathbf{X}_t(\mathbf{u}_t - \boldsymbol{\alpha}_t^{(h)})\|_{M_t}^2 , \tag{21}$$

*for* $\mathbf{u}_t \in \mathbb{R}^{n_t}$ *with*

$$\mathbf{u}_t^i \in \partial \ell_t(\mathbf{w}_t(\boldsymbol{\alpha})^\top \mathbf{x}_t^i, y_t^i) . \tag{22}$$

*Proof.* Applying Lemma 4 and recalling $\mathcal{D}(\boldsymbol{\alpha}) = \sum_{t=1}^m \mathcal{G}_t^{\sigma'}(\mathbf{0}; \mathbf{v}, \boldsymbol{\alpha}_t)$, we can first bound the improvement for each task separately. Following a similar approach as in the proof of [49, Lemma 7] we can obtain the bound (19) which bounds the improvement from $\boldsymbol{\alpha}^{(h)}$ to $\boldsymbol{\alpha}^{(h+1)}$. $\qquad \square$

The following lemma relates the improvement of the dual objective in one iteration to the duality gap for the smooth loss functions $\ell_t$.

**Lemma 6.** *If the loss functions $\ell_t$ are $(1/\mu)$-smooth, then there exists a proper constants $s \in (0, 1]$, such that for any $\gamma \in (0, 1]$ at any iteration $h$*

$$\mathbb{E}\left[ \mathcal{D}(\boldsymbol{\alpha}^{(h)}) - \mathcal{D}(\boldsymbol{\alpha}^{(h+1)}) | \mathcal{H}_h \right] \geq s\gamma(1 - \bar{\Theta}) G(\boldsymbol{\alpha}^{(h)}), \tag{23}$$

*where $G(\boldsymbol{\alpha}^{(h)})$ is the duality gap of $\boldsymbol{\alpha}^{(h)}$ which is defined in (17).*

*Proof.* Recall the definition of $\sigma_{\max}$ in (16). Now, if we carefully choose $s = \mu/(\mu + \sigma_{\max}\sigma')$, it is easy to show that $J_t \leq 0$ in (19); see [49, Theorem 11] for details. The final result follows as a consequence of Lemma 5. $\qquad \square$

Note that Lemma 5 holds even if the functions are non-smooth, i.e. $\mu = 0$. However, we cannot infer sufficient decrease of Lemma 6 from Lemma 5 when $\mu = 0$. Therefore, we need additional tools when the losses are $L$-Liptchitz. The first is the following lemma, which bounds the $J$ term in (19).

**Lemma 7.** *Assume that the loss functions $\ell_t$ are $L$-Lipschitz. Denote $J := \sum_{t=1}^m J_t$, where $J_t$ is defined in (21), then*

$$J \leq 4L^2 \sum_{t=1}^m \sigma_t n_t := 4L^2 \sigma, \tag{24}$$

*where $\sigma_t$ is defined in (16).*

*Proof.* The proof is similar to [31, Lemma 6] and using the definitions of $\sigma$ and $\sigma_t$ and the fact that the losses are $L$-Lipchitz. $\qquad \square$

## C.1 Convergence Analysis for Smooth Losses

### C.1.1 Proof of Theorem 1

Let us rewrite (23) from Lemma 6 as

$$\mathbb{E}[\mathcal{D}(\boldsymbol{\alpha}^{(h)}) - \mathcal{D}(\boldsymbol{\alpha}^{(h+1)}) | \mathcal{H}_h] = \mathcal{D}(\boldsymbol{\alpha}^{(h)}) - \mathcal{D}(\boldsymbol{\alpha}^\star) + \mathbb{E}[\mathcal{D}(\boldsymbol{\alpha}^\star) - \mathcal{D}(\boldsymbol{\alpha}^{(h+1)}) | \mathcal{H}_h]$$
$$\geq s\gamma(1 - \bar{\Theta}) G(\boldsymbol{\alpha}^{(h)})$$
$$\geq s\gamma(1 - \bar{\Theta}) \left( \mathcal{D}(\boldsymbol{\alpha}^{(h)}) - \mathcal{D}(\boldsymbol{\alpha}^\star) \right),$$

where the last inequality is due to weak duality, i.e. $G(\boldsymbol{\alpha}^{(h)}) \geq \mathcal{D}(\boldsymbol{\alpha}^{(h)}) - \mathcal{D}(\boldsymbol{\alpha}^\star)$. Re-arranging the terms in the above inequality, we can easily get

$$\mathbb{E}[\mathcal{D}(\boldsymbol{\alpha}^{(h+1)}) - \mathcal{D}(\boldsymbol{\alpha}^\star) | \mathcal{H}_h] \leq \left( 1 - s\gamma(1 - \bar{\Theta}) \right) \left( \mathcal{D}(\boldsymbol{\alpha}^{(h)}) - \mathcal{D}(\boldsymbol{\alpha}^\star) \right) \tag{25}$$

Recursively applying this inequality and taking expectations from both sides, we arrive at

$$\mathbb{E}[\mathcal{D}(\boldsymbol{\alpha}^{(h+1)}) - \mathcal{D}(\boldsymbol{\alpha}^\star)] \leq \left(1 - s\gamma(1 - \bar{\Theta})\right)^{h+1} \left(\mathcal{D}(\boldsymbol{\alpha}^{(0)}) - \mathcal{D}(\boldsymbol{\alpha}^\star)\right). \tag{26}$$

Now we can use a simple bound on the initial duality gap [49, Lemma 10], which states that $\mathcal{D}(\boldsymbol{\alpha}^{(0)}) - \mathcal{D}(\boldsymbol{\alpha}^\star) \leq n$, to get the final result. It is worth noting that we can translate the bound on the dual distance to optimality to the bound on the duality gap using the following inequalities

$$s\gamma(1 - \bar{\Theta})\,\mathbb{E}[G(\boldsymbol{\alpha}^{(H)})] \leq \mathbb{E}[\mathcal{D}(\boldsymbol{\alpha}^{(H)}) - \mathcal{D}(\boldsymbol{\alpha}^{(H+1)})] \leq \mathbb{E}[\mathcal{D}(\boldsymbol{\alpha}^{(H)}) - \mathcal{D}(\boldsymbol{\alpha}^\star)] \leq \epsilon_{\mathcal{D}}, \tag{27}$$

where the first inequality is due to (23), the second inequality is due to the optimality of $\boldsymbol{\alpha}^\star$, and the last inequality is the bound we just proved for the dual distance to optimality.

### C.1.2 Asymptotic Convergence

In the case of smooth loss functions, it is possible to get asymptotic convergence results under milder assumptions. The following corollary is an extension of Theorem 1.

**Corollary 8.** *If the loss functions $\ell_t$ are $\mu$-smooth, then under Assumption 1, $\mathbb{E}[\mathcal{D}(\boldsymbol{\alpha}^{(H)}) - \mathcal{D}(\boldsymbol{\alpha}^\star)] \to 0$ as $H \to \infty$ if either of the following conditions hold*

- *$\limsup_{h\to\infty} p_t^h < 1$ and $\limsup_{h\to\infty} \hat{\Theta}_t^h < 1$.*

- *For any task $t$, $\left(1 - p_t^h\right) \times \left(1 - \hat{\Theta}_t^h\right) = \omega(\frac{1}{h})$. Note that in this case $\lim_{h\to\infty} p_t^h$ can be equal to 1.*

*Proof.* The proof is similar to the proof of Theorem 1. We can use the same steps to get a sufficient decrease inequality like the one in (25), with $\bar{\Theta}$ replaced with $\bar{\Theta}^h := \max_t \Theta_t^h$.

$$\mathbb{E}[\mathcal{D}(\boldsymbol{\alpha}^{(h+1)}) - \mathcal{D}(\boldsymbol{\alpha}^\star)|\mathcal{H}_h] \leq \left(1 - s\gamma(1 - \bar{\Theta}^h)\right)\left(\mathcal{D}(\boldsymbol{\alpha}^{(h)}) - \mathcal{D}(\boldsymbol{\alpha}^\star)\right)$$

The rest of the argument follows by applying this inequality recursively and using the assumptions in the corollary. $\qquad\square$

### C.2 Convergence Analysis for Lipschitz Losses: Proof for Theorem 2

*Proof.* For $L$-Lipschitz loss functions, the proof follows the same line of reasoning as the proof of Theorem 8 in [31] and therefore we do not cover it in detail. Unlike the case with smooth losses, it is not possible to bound the decrease in dual objective by (23). However, we can use Lemma 5 with $\mu = 0$. The next step is to bound $J = \sum_{t=1}^m J_t$ in (19), which can be done via Lemma 7. Finally, we apply the inequalities recursively, choose $s$ carefully, and bound the terms in the final inequality. We refer the reader to the proof of Theorem 8 in [31] for more details. It is worth noting that similar to Theorem 1, we can similarly get bounds on the expected duality gap, instead of the dual objective. $\qquad\square$

## D   Choosing $\sigma'$

In order to guarantee the convergence of the federated update of MOCHA, the parameter $\sigma'$ must satisfy:

$$\sigma' \sum_{t=1}^m \|\mathbf{X}_t \boldsymbol{\alpha}_t\|_{\mathbf{M}_t}^2 \geq \gamma \|\mathbf{X}\boldsymbol{\alpha}\|_{\mathbf{M}}^2 \quad \forall \boldsymbol{\alpha} \in \mathbb{R}^n, \tag{28}$$

where $\gamma \in (0, 1]$ is the aggregation parameter for MOCHA Algorithm. Note that in Algorithm 1 we have assumed that $\gamma = 1$. Based on Remark 1, it can be seen that the matrix $\mathbf{M}$ in Assumption 1 can be chosen of the form $\mathbf{M} = \bar{\mathbf{M}} \otimes \mathbf{I}_{d\times d}$, where $\bar{\mathbf{M}}$ is a positive definite matrix of size $m \times m$. For such a matrix, the following lemma shows how to choose $\sigma'$.

**Lemma 9.** *For any positive definite matrix $\mathbf{M} = \bar{\mathbf{M}} \otimes \mathbf{I}_{d\times d}$,*

$$\sigma' := \gamma \max_t \sum_{t'=1}^m \frac{|\bar{\mathbf{M}}_{tt'}|}{\bar{\mathbf{M}}_{tt}} \tag{29}$$

*satisfies the inequality* (28).

*Proof.* First of all it is worth noting that for any $t$, $\mathbf{M}_t = \bar{\mathbf{M}}_t \otimes \mathbf{I}_{d \times d}$. For any $\boldsymbol{\alpha} \in \mathbb{R}^n$

$$\gamma \|\mathbf{X}\boldsymbol{\alpha}\|_{\mathbf{M}}^2 = \gamma \sum_{t,t'} \bar{\mathbf{M}}_{tt'} \langle \mathbf{X}_t \boldsymbol{\alpha}_t, \mathbf{X}_{t'} \boldsymbol{\alpha}_{t'} \rangle$$

$$\leq \gamma \sum_{t,t'} \frac{1}{2} |\bar{\mathbf{M}}_{tt'}| \left( \frac{1}{\bar{\mathbf{M}}_{tt}} \|\mathbf{X}_t \boldsymbol{\alpha}_t\|_{\mathbf{M}_t}^2 + \frac{1}{\bar{\mathbf{M}}_{t't'}} \|\mathbf{X}_{t'} \boldsymbol{\alpha}_{t'}\|_{\mathbf{M}_{t'}}^2 \right)$$

$$= \gamma \sum_t \left( \sum_{t'} \frac{|\bar{\mathbf{M}}_{tt'}|}{\bar{\mathbf{M}}_{tt}} \right) \|\mathbf{X}_t \boldsymbol{\alpha}_t\|_{\mathbf{M}_t}^2$$

$$\leq \sigma' \sum_t \|\mathbf{X}_t \boldsymbol{\alpha}_t\|_{\mathbf{M}_t}^2,$$

where the first inequality is due to Cauchy-Schwartz and the second inequality is due to definition of $\sigma'$. $\qquad\square$

**Remark 5.** *Based on the proof of Lemma 9, it is easy to see that we can choose $\sigma'$ differently across the tasks in our algorithm to allow tasks that are more loosely correlated with other tasks to update more aggressively. To be more specific, if we choose $\sigma'_t = \gamma \sum_{t'} \frac{|\bar{\mathbf{M}}_{tt'}|}{\bar{\mathbf{M}}_{tt}}$, then it it is possible to show that $\gamma \|\mathbf{X}\boldsymbol{\alpha}\|_{\mathbf{M}}^2 \leq \sum_{t=1}^m \sigma'_t \|\mathbf{X}_t \boldsymbol{\alpha}_t\|_{\mathbf{M}_t}^2$ for any $\boldsymbol{\alpha}$, and the rest of the convergence proofs will follow.*

### D.1 The Role of Aggregation Parameter $\gamma$

The following remark highlights the role of aggregation parameter $\gamma$.

**Remark 6.** *Note that the when $\gamma < 1$ the chosen $\sigma'$ in (28) would be smaller compared to the case where $\gamma = 1$. This means that the local subproblems would be solved with less restrictive regularizer. Therefore, the resulting $\Delta\boldsymbol{\alpha}$ would be more aggressive. As a result, we need to do a more conservative update $\boldsymbol{\alpha} + \gamma\Delta\boldsymbol{\alpha}$ in order to guarantee the convergence.*

Although aggregation parameter $\gamma$ is proposed to capture this trade off between aggressive subproblems and conservative updates, in most practical scenarios $\gamma = 1$ has the best empirical performance.

## E  Simulation Details

In this section, we provide additional details and results of our empirical study.

### E.1 Datasets

In Table 2, we provide details on the number of tasks ($m$), feature size ($d$), and per-task data size ($n_t$) for each federated dataset described in Section 5. The standard deviation $n_\sigma$ is a measure data skew, and calculates the deviation in the sizes of training data points for each task, $n_t$. All datasets are publicly available.

Table 2: Federated Datasets for Empirical Study.

| Dataset | Tasks ($m$) | Features ($d$) | Min $n_t$ | Max $n_t$ | Std. Deviation $n_\sigma$ |
|---------|-------------|----------------|-----------|-----------|---------------------------|
| Human Activity | 30 | 561 | 210 | 306 | 26.75 |
| Google Glass | 38 | 180 | 524 | 581 | 11.07 |
| Vehicle Sensor | 23 | 100 | 872 | 1,933 | 267.47 |

### E.2 Multi-Task Learning with Highly Skewed Data

To generate highly skewed data, we sample from the original training datasets so that the task dataset sizes differ by at least two orders of magnitude. The sizes of these highly skewed datasets are shown in Table 3. When looking at the performance of local, global, and multi-task models for these datasets

Table 3: Skewed Datasets for Empirical Study.

| Dataset | Tasks ($m$) | Features ($d$) | Min $n_t$ | Max $n_t$ | Std. Deviation $\sigma$ |
|---------|-------------|----------------|-----------|-----------|-------------------------|
| HA-Skew | 30 | 561 | 3 | 306 | 84.41 |
| GG-Skew | 38 | 180 | 6 | 581 | 111.79 |
| VS-Skew | 23 | 100 | 19 | 1,933 | 486.08 |

(Table 4), the global model performs slightly better in this setting (particularly for the Human Activity dataset). However, multi-task learning still significantly outperforms all models.

Table 4: Average prediction error for skewed data: means and standard errors over 10 random shuffles.

| Model | HA-Skew | GG-Skew | VS-Skew |
|-------|---------|---------|---------|
| Global | 2.41 (0.30) | 5.38 (0.26) | 13.58 (0.23) |
| Local | 3.87 (0.37) | 4.96 (0.20) | 8.15 (0.19) |
| MTL | **1.93 (0.44)** | **3.28 (0.15)** | **6.91 (0.21)** |

### E.3 Implementation Details

In this section, we provide details on our experimental setup and compared methods.

**Methods.**

- **Mb-SGD.** Mini-batch stochastic gradient descent is a standard, widely used method for parallel and distributed optimization. See, e.g., a discussion of this method for the SVM models of interest [46]. We tune both the mini-batch size and step size for best performance using grid search.

- **Mb-SDCA.** Mini-batch SDCA aims to improve mini-batch SGD by employing coordinate ascent in the dual, which has encouraging theoretical and practical backings [47, 50]. For all experiments, we scale the updates for mini-batch stochastic dual coordinate ascent at each round by $\frac{\beta}{b}$ for mini-batch size $b$ and $\beta \in [1, b]$, and tune both parameters with grid search.

- **COCOA.** We generalize COCOA [22, 31] to solve (1), and tune $\theta$, the fixed approximation parameter, between $[0, 1)$ via grid search. For both COCOA, and MOCHA, we use coordinate ascent as a local solver for the dual subproblems (4).

- **MOCHA.** The only parameter necessary to tune for MOCHA is the level of approximation quality $\theta_t^h$, which can be directly tuned via $H_i$, the number of local iterations of the iterative method run locally. In Section 4, our theory relates this parameter to global convergence, and we discuss the practical effects of this parameter in Section 3.4.

**Computation and Communication Complexities.** We provide a brief summary of the above methods from the point of view of computation, communication, and memory complexities. MOCHA is superior in terms of its computation complexity compared to other distributed optimization methods, as MOCHA allows for flexibility in its update of W. At one extreme, the update can be based on a single data point per iteration in parallel, similar to parallel SGD. At the other extreme, MOCHA can completely solve the subproblems on each machine, similar to methods such as ADMM. This flexibility of computation yields direct benefits in terms of communication complexity, as performing additional local computation will result in fewer communication steps. Note that all methods, including MOCHA, communicate the same size vector at each iteration, and so the main difference is in how many communication rounds are necessary for convergence. In terms of memory, MOCHA must maintain the task matrix, $\Omega$, on the master server. While this overhead is greater than most *non-MTL* (global or local) approaches, the task matrix is typically low-rank by design and the overhead is thus manageable. We discuss methods for computing $\Omega$ in further detail in Section B.3.

**Estimated Time.**   To estimate the time to run methods in the federated setting, we carefully count the floating-point operations (FLOPs) performed in each local iteration for each method, as well as the size and frequency of communication. We convert these counts to estimated time (in milliseconds), using known clock rate and bandwidth/latency numbers for mobile phones in 3G, LTE, and wireless networks [52, 20, 48, 9, 38]. In particular, we use the following standard model for the cost of one round, $h$, of local computation / communication on a node $t$:

$$Time(h,t) := \frac{FLOPs(h,t)}{Clock\ Rate(t)} + Comm(h,t) \tag{30}$$

Note that the communication cost $Comm(h,t)$ includes both bandwidth and latency measures. Detailed models of this type have been used to closely match the performance of real-world systems [40].

**Statistical Heterogeneity.**   To account for statistical heterogeneity, MOCHA and the mini-batch methods (Mb-SGD and Mb-SDCA) can adjust the number of local iterations or batch size, respectively, to account for difficult local problems or high data skew. However, because COCOA uses a fixed accuracy parameter $\theta$ across both the tasks and rounds, changes in the subproblem difficulty and data skew can make the computation on some nodes much slower than on others. For COCOA, we compute $\theta$ via the duality gap, and carefully tune this parameter between $[0,1)$ for best performance. Despite this, the number of local iterations needed for $\theta$ varies significantly across nodes, and as the method runs, the iterations tend to increase as the subproblems become more difficult.

**Systems Heterogeneity.**   Beyond statistical heterogeneity, there can be variability in the systems themselves that cause changes in performance. For example, low battery levels, poor network connections, or low memory may reduce the ability a solver has on a local node to compute updates. As discussed in Section 3.4, MOCHA assumes that the central node sets some global clock cycle, and the $t$-th worker determines the amount of feasible local computation given this clock cycle along with its systems constraints. This specified amount of local computation corresponds to some implicit value of $\theta_t^h$ based on the underlying systems and statistical challenges for the $t$-th node.

To model this setup in our simulations, it suffices to fix a global clock sycle and then randomly assign various amounts of local computation to each local node at each iteration. Specifically, in our simulations we charge all nodes the same fixed computation cost at each iteration over an LTE network, but force some nodes to perform less updates given their current systems constraints. At each round, we assign the number of updates for node $t$ between $[0.1n_{\min}, n_{\min}]$ for *high variability* environments, and between $[0.9n_{\min}, n_{\min}]$ for *low variability* environments, where $n_{\min} := \min_t n_t$ is the minimum number of local data points across tasks.

For the mini-batch methods, we vary the mini-batch size in a similar fashion. However, we do not follow this same process for COCOA, as this would require making the $\theta$ parameter worse than what was optimally tuned given statistical heterogeneity. Hence, in these simulations we do not introduce any additional variability (and thus present overly optimistic results for COCOA). In spite of this, we see that in both low and high variability settings, MOCHA significantly outperforms all other methods and is robust to systems-related heterogeneity.

**Fault Tolerance.**   Finally, we demonstrate that MOCHA can handle nodes periodically dropping out, which is also supported in our convergence results in Section 4. We perform this simulation using the notation defined in Assumption 2, i.e., that each node $t$ temporarily drops on iteration $h$ with probability $p_t^h$. In our simulations, we modify this probability directly and show that MOCHA is robust to fault tolerance in Figure 3. However, note that this robustness is not merely due to statistical redundancy: If we are to drop out a node entirely (as shown in the green dotted line), MOCHA will not converge to the correct solution. This provides insight into our Assumption 2, which requires that the probability that a node drops at each round cannot be exactly equal to one.