[Reviews · NeurIPS 2017]

Reviewer 1



This work extends on both federated learning and MTL to present the federated MTL, which considers important practical issues like reliability of computing devices and the communication channel. Overall it’s a well written paper. The challenges and contributions are clearly presented. The proposed algorithm should run reasonably efficiently (although it would be better if the authors can further list the computing and memory complexity overhead, if any, compared to previous work, in the supplementary material). The theoretical analyses are complete and correct as far as I can tell. The experiments are comprehensive and the results are indeed impressive. The only major suggestion I would give is to possibly include discussion (or future work) addressing the limitation of requiring convex loss in real applications, how techniques presented in this work may benefit distributed nonconvex problems, and/or how they may benefit from “convexified” stronger models (e.g. [1-4]). [1] Convex Deep Learning via Normalized Kernels, NIPS 2014 [2] Convolutional Kernel Networks, NIPS 2014 [3] Tensor Switching Networks, NIPS 2016 [4] Convexified Convolutional Neural Networks, ICML 2017

Reviewer 2



The paper generalizes an existing framework for distributed multitask learning (COCOA) and extends it to handle practical systems challenges such as communications cost, stragglers, etc. The proposed technique (MOCHA) makes the framework robust and faster. Pros: - Rigorous theoretical analysis - Sound experiment methodology Cons: - Not very novel (compared to [47]) and just a mild variation over [47] including the analysis. Major Comments: --------------- 1. The Equation 5 depends on \Delta \alpha_t* (minimizer of sub-problem). My understanding is that this minimizer is not known and cannot be computed accurately at every node, and therefore, we need the approximation. If so, then how do the nodes set \theta^h_t? Appendix E.3 states that this parameter is directly tuned via H_i. However, the quality should be determined by the environment, and not supposed to be 'tuned'. If tuned, then it might act as a 'learning rate' -- instead of taking the full step in the direction of the gradient, the algorithm takes a partial step. 2. The main difference between this paper and [47] seems to be that instead of the same \theta in [47], this paper has defined separate \theta^h_t for each node t. Minor comments / typos: ----------------------- 39: ...that MTL is -as- a natural .... 108: ...of this (12) in our... -- should be (2)

Reviewer 3



This work proposes a refinement of multi-task learning for systems that have bandwidth, reliability, and battery constraints (federated learning). The extension of CoCoA seem rather direct. The dual, as most duals in this type of optimization problem, is amenable to a distributed computation framework. The change in regularization of the primal is a minor, but important point. After the author feedback, it became clear that the synchronous updates and the changes to CoCoa have merit. Most of the intellectual contribution is on the convergence analysis rather than the learning algorithm, which is both a strong and a weak point. Overall the paper seems like a good contribution.